# Analysis of inter-hospital transfer on clinical outcomes after primary percutaneous coronary intervention for ST-segment elevation myocardial infarction: A secondary analysis of the BRIGHT-4 trial

Xiaolin Su[1,2☯], Miaohan Qiu[1☯], Chengqi Gu[3], Xiuhui Yang[4], Bin Liu[5], Fanbo Meng[6], Bin Ning[7], Wei Li[8], Zhixiong Zhong[9], Zhengzhong Wang[10], Bei Shi[11], Zhuo Shang[12], Zhenyang Liang[1], Yi Li[1], Yaling Han[1*], Gregg W. Stone[13]

1 State Key Laboratory of Frigid Zone Cardiovascular Diseases, Department of Cardiology, General Hospital of Northern Theatre Command, Shenyang, China, 2 Department of Traditional Chinese Medicine, Shengjing Hospital of China Medical University, Shenyang, China, 3 Datong Third People's Hospital, Datong, China, 4 Luohe Central Hospital, Luohe, China, 5 The Second Hospital of Jilin University, Changchun, China, 6 China-Japan Union Hospital of Jilin University, Changchun, China, 7 Fuyang People's Hospital, Fuyang, China, 8 The Affiliated Hospital of Guizhou Medical University, Guiyang, China, 9 Meizhou People's Hospital, Meizhou, China, 10 Qingdao Municipal Hospital, Qingdao, China, 11 Affiliated Hospital of Zunyi Medical University, Zunyi, China, 12 Bengbu Second People's Hospital, Bengbu, China, 13 Zena and Michael A. Wiener Cardiovascular Institute, Icahn School of Medicine at Mount Sinai, New York, New York, United States of America

☯ These authors contributed equally to this work.
* hanyaling@163.net

## Abstract

### Background

Previous studies evaluating the influence of inter-hospital transfer on mortality in ST-segment elevation myocardial infarction (STEMI) patients undergoing primary percutaneous coronary intervention (PCI) reported conflicting results. The multicenter BRIGHT-4 trial demonstrated that bivalirudin plus a post-PCI high-dose infusion (1.75 mg/kg/h) reduced the 30-day primary endpoint of all-cause mortality or Bleeding Academic Research Consortium (BARC) types 3–5 bleeding compared with heparin monotherapy in STEMI patients. This study aimed to assess the impact of inter-hospital transfer on clinical outcomes and the effectiveness of bivalirudin versus heparin in STEMI patients undergoing PCI.

### Methods and findings

In BRIGHT-4, 2,121 (35.7%) patients were transferred to a tertiary hospital for primary PCI while 3,817 (64.3%) were directly admitted to an interventional facility. The primary outcome was the composite of all-cause death or BARC types 3–5 bleeding occurring within 30 days. The secondary outcomes included stent thrombosis.

**Data availability statement:** The authors are willing to share de-identified individual data with researchers who provide a methodologically sound proposal. Interested parties should contact the Clinical Research Data Center of the General Hospital of Northern Theatre Command via Lcyjsjzx@163.com.

**Funding:** The BRIGHT-4 trial was funded by the Chinese Society of Cardiology Foundation (CSCF2019A01, granted to YH), the National Key Research and Development Project (2022YFC2503500, granted to YH; 2022YFC2503504, granted to YL), and a research grant from Jiangsu Hengrui Pharmaceuticals (granted to YH). The funders had no role in study design, data collection and analysis, decision to publish, or preparation of the manuscript. None of the authors received salary support from any of the funders.

**Competing interests:** I have read the journal's policy and the authors of this manuscript have the following competing interests: GWS has received speaker honoraria from Pulnovo, Medtronic, Amgen, Boehringer Ingelheim, Abiomed; has served as a consultant to CorFlow, Cardiomech, Robocath, Daiichi Sankyo, Ablative Solutions, Vectorious, Miracor, Apollo Therapeutics, Elucid Bio, Abbott, Cardiac Success, Occlutech, Millennia Biopharma, Remote Cardiac Enablement, Valfix, Zoll, HeartFlow, Shockwave, Impulse Dynamics, Adona Medical, Oxitope, HighLife, Elixir, Aria; and has equity/options from Cardiac Success, Ancora, Cagent, Applied Therapeutics, Biostar family of funds, SpectraWave, Orchestra Biomed, Aria, Valfix, Xenter. GWS's employer, Mount Sinai Hospital, receives research grants from Shockwave, Biosense-Webster, Abbott, Abiomed, Bioventrix, Cardiovascular Systems Inc, Phillips, Vascular Dynamics, Pulnovo, V-wave and PCORI (via Weill Cornell Medical Center). The other authors have no disclosures.

**Abbreviations:** DIDO, door-in door-out; FMC, first medical contact; PCI, percutaneous coronary intervention; PSM, propensity score matching; STEMI, ST-segment elevation myocardial infarction.

Adjustments were made for baseline covariates and randomized treatments. Transferred patients had a longer median time from symptom onset to wire crossing the infarct-related artery (6.00 versus 3.93 hrs, $P < 0.0001$). At 30 days, there were no significant between-group differences in the rates of the primary outcome (4.2% versus 3.4%, adjusted hazard ratio [HR] 0.99, 95% confidence intervals [CI] 0.73, 1.33, $P = 0.94$) or its components. Bivalirudin with a high-dose post-PCI infusion was associated with consistent reductions of the primary outcome in the transfer (3.5% versus 4.8%, adjusted HR 0.66, 95%CI 0.42, 1.05) and direct admission (2.8% versus 4.1%, adjusted HR 0.62, 95% CI 0.43, 0.89) group compared with heparin monotherapy ($P_{interaction} = 0.78$), as well as individually for stent thrombosis. The main limitations of this study are that it is a *post hoc* analysis, and the long-term prognostic impact of transfer on STEMI patients requires further investigation.

## Conclusions

In this *post hoc* analysis, 30-day clinical outcomes for STEMI patients transferred for PCI were not significantly worse than direct admission patients. Bivalirudin with a post-PCI high-dose infusion for 2–4 hrs was associated with lower rates of 30-day all-cause mortality, major bleeding and stent thrombosis, consistently observed in transfer and direct admission patients.

## Trial registration

BRIGHT-4 trial NCT03822975 http://www.clinicaltrials.gov

## Author summary

### Why was this study done?

- Many ST-segment elevation myocardial infarction (STEMI) patients present at non-percutaneous coronary intervention (PCI) capable hospitals and require transfer, a process that may prolong ischemic time and increase mortality.

- The large-scale BRIGHT-4 trial found that using bivalirudin instead of heparin, followed by a 2–4-hr post-PCI high-dose infusion, significantly lowered the risk of death or major bleeding within 30 days in patients with STEMI undergoing primary PCI.

- Current clinical practice lacks evidence on outcomes and optimal anticoagulation for transferred STEMI patients.

### What did the researchers do and find?

- We conducted a *post hoc* analysis of the BRIGHT-4 trial, focusing on 5938 patients with STEMI undergoing primary PCI, all of whom had clear records of how they arrived at PCI-capable hospitals.

- Despite a 2-hr delay to reperfusion, patients who had to be transferred for primary PCI did not have worse outcomes after 30 days compared to those who arrived directly at PCI-capable hospitals.

- The association between bivalirudin use and lower rates of death or major bleeding appeared to be maintained regardless of the need for inter-hospital transfer.

### What do these findings mean?

- This study emphasized that speed mattered most - timely PCI treatment was more important than how patients got to the hospital, whether they came directly or were transferred.

- When treating STEMI patients with primary PCI, bivalirudin could be a reasonable option no matter how the patient got to the hospital – even if they had to be moved from another facility. The connection between using bivalirudin and better outcomes (fewer deaths, less bleeding) stayed consistent either way.

- The generalizability of the present study is limited by not fully balancing the differences between patients who were transferred and those who arrived directly at the hospital, as well as by a relatively short follow-up period. More research is needed to see how inter-hospital transfer affects the long-term prognosis of STEMI patients and to assess the benefits of bivalirudin over time.

## Background

Primary percutaneous coronary intervention (PCI) is the preferred method to achieve reperfusion in patients with ST-segment elevation myocardial infarction (STEMI), provided that it can be performed in a timely fashion [1,2]. However, approximately 20%−50% of STEMI patients require transfer to a PCI center after initial presentation to a non-PCI capable hospital [3–6], resulting in longer ischemic times compared with direct admission to tertiary interventional hospitals. As a result, more than 40% of transferred patients in the United States fail to achieve the guideline-recommended first medical contact (FMC)-to-device time of 120 min [7], which may reduce myocardial salvage. In some studies, inter-hospital transfer for primary PCI has been identified as a major predictor of higher 12-month mortality [4], whereas other studies have not demonstrated clearly worsened outcomes after transfer for primary PCI [6,8–11].

Prolonged ischemic time has been associated with increased fibrin content of intracoronary thrombi [12], making them more resistant to anticoagulation with heparin which binds primarily with fluid-phase but not clot-bound thrombus. Bivalirudin, in contrast, has high affinity for both clot-bound and fluid-phase thrombin, preventing the initiation and propagation of clot formation [13]. Bivalirudin might thus be a more appropriate anticoagulation strategy for patients presenting late after symptom onset, including those undergoing inter-hospital transfer. In the HORIZONS-AMI trial, procedural anticoagulation with bivalirudin resulted in lower rates of cardiac mortality and major bleeding compared with heparin plus a glycoprotein IIb/IIIa inhibitor (GPI), regardless of transfer for primary PCI [6]. However, in current clinical practice in which GPI are not routinely used with heparin and a post-PCI bivalirudin infusion is recommended after primary PCI, little is known regarding the outcomes of and the optimal anticoagulation strategy for transferred patients with STEMI.

In the large-scale BRIGHT-4 trial, bivalirudin plus a post-PCI high-dose infusion (1.75 mg/kg/h) for 2–4 hrs reduced the 30-day rates of all-cause mortality, major bleeding and stent thrombosis compared with heparin monotherapy in patients with STEMI undergoing primary PCI [14]. The present *post hoc* analysis therefore aimed to compare the clinical outcomes of patients with STEMI transferred to tertiary hospitals for PCI with those who presented directly at an interventional facility, and to investigate whether inter-hospital transfer had an impact on the outcomes of anticoagulation with bivalirudin versus heparin as used in contemporary clinical practice.

## Methods

### Study design and participants

The design, major inclusion and exclusion criteria, and results of the BRIGHT-4 trial have been published [14,15]. Briefly, BRIGHT-4 was an investigator-sponsored, open-label, randomized controlled trial conducted at 87 clinical centers in China. Between February 14th, 2019 and April 7th, 2022, 6,016 patients with STEMI undergoing primary PCI with radial artery access within 48 hrs of symptom onset were enrolled. Exclusion criteria included use of fibrinolytic therapy, anticoagulants or GPI before cardiac catheterization. Eligible patients were randomly assigned (1:1) to receive either bivalirudin with a post-PCI high-dose infusion for 2–4 hrs or unfractionated heparin monotherapy. The use of GPI was reserved for procedural thrombotic complications in both groups. The study protocol was approved by the ethics committee of the General Hospital of Northern Theatre Command as well as at each participating center (S1 Text). The study was performed in accordance with the principles of the Declaration of Helsinki. Written informed consent was provided by all patients or their legal representatives before randomization. Further details regarding the original study protocol and statistical analysis plan are provided in the Supporting Information (S2 and S3 Texts). This study is reported as per the Consolidated Standards of Reporting Trials (CONSORT) guideline (S1 Checklist).

For the present *post hoc* analysis, 78 (1.3%) patients with uncertain mode of arrival at the PCI capable hospital were excluded. The remaining 5,938 patients were categorized into two groups: (1) patients who initially presented at a tertiary hospital for primary PCI (the direct admission group), and (2) patients who initially presented at a non-PCI capable hospital before being transferred to a tertiary facility for PCI (inter-hospital transfer group). All patients were randomized at the tertiary facility before receiving any study drug. All patients also received dual antiplatelet therapy with aspirin and either clopidogrel or ticagrelor at physician discretion. Other medications were prescribed according to current guidelines.

### Clinical outcomes and definitions

The primary outcome was the composite of all-cause death or Bleeding Academic Research Consortium (BARC) types 3–5 bleeding occurring within 30 days after randomization. The secondary outcomes were major adverse cardiac or cerebral events (MACCE, defined as the composite of all-cause death, (recurrent) myocardial infarction (MI), ischemia-driven target vessel revascularization (TVR), or stroke) and its components; stent thrombosis according to Academic Research Consortium definite or probable criteria; BARC types 2–5 bleeding; the composite of all-cause death or BARC types 2–5 bleeding; acquired thrombocytopenia; and net adverse clinical events (NACE, the composite of MACCE or BARC types 3–5 bleeding). The specific definitions for these outcomes have been previously described [14]. All endpoints were adjudicated by an independent committee using original source documents.

Symptom onset was the time that symptoms (e.g., chest pain, dyspnea, cardiac arrest) began that led the patient to seek medical attention [16]. FMC was the time when the paramedics arrived at the patient's side. The times of arrival at the transfer and tertiary hospitals were recorded. Wire time was the time of wire crossing the infarct lesion [2].

### Statistical analysis

Continuous variables are presented as mean and standard deviation (SD) or median and interquartile ranges (IQR) and were compared using a Student $t$ test or the Wilcoxon rank-sum test respectively. Categorical variables are expressed as frequencies (%) and were compared using the $\chi^2$ test or Fisher's exact test. No imputation was used to infer missing values. Time-to-event rates were estimated using the Kaplan–Meier method and were compared with the log-rank test.

Multivariable Cox proportional hazards regression models were used to adjust baseline imbalances between groups. Variables included in the Cox model were age, sex, body mass index, hypertension, diabetes, smoking, previous MI, previous PCI, previous stroke, Killip class, hemoglobin, platelet count, estimated glomerular filtration rate (eGFR), medical therapy (randomized anticoagulant, aspirin, P2Y12 inhibitors, bail-out tirofiban), arterial access, coronary arteries treated,

thrombus aspiration, pre- and post-PCI TIMI flow, and procedure strategies (PCI, coronary artery bypass surgery, coronary angiography, none). The Cox proportional hazards assumption for the primary outcome was confirmed graphically using log(−log) plots. Given the multi-center design of the study, Cox frailty models were further applied to estimate hazard ratios (HRs) for the primary outcome, incorporating center as a random intercept to account for clustering effects.

To confirm the robustness of our findings, propensity score matching (PSM) was conducted between patients with and without inter-hospital transfer using a 1:1 nearest-neighbor matching algorithm with a caliper of 0.05 standard deviations. The matched variables were identical to those included as covariates in multivariable Cox models. In addition, we excluded patients with symptom onset-to-wire time exceeding 24 hrs and employed multivariable Cox models to test the consistency of our hypothesis. All statistical analyses were two-sided and were performed with SAS version 9.4.

## Results

### Baseline characteristics

Among the 5,938 study patients, 2,121 (35.7%) were transferred to the tertiary hospital for primary PCI while 3,817 (64.3%) patients were directly admitted to a PCI-capable facility. The flowchart of the study is provided in Fig 1. Patients in the inter-hospital transfer group were less likely to have hypertension and history of MI or PCI compared with those in the direct admission group but were more likely to smoke and have anemia (Table 1). Symptom onset-to-FMC time was significantly shorter in the inter-hospital transfer group as was the duration from tertiary hospital arrival to wire time. However, after accounting for a median inter-hospital transfer time of 132 min, the median symptom onset-to-wire time was approximately 2 hrs longer in the inter-hospital transfer group compared with the direct admission group (6.00 versus 3.93 hrs, $P < 0.0001$) (Table 1 and Fig 2).

Procedural data and medications are presented in Tables 2 and S1. Compared with the direct admission group, the patients requiring inter-hospital transfer were more likely to be treated with clopidogrel than ticagrelor, had a slightly higher use of radial artery access, were slightly less likely to undergo PCI (but more often in the left anterior descending than the right coronary artery), were less likely to undergo thrombus aspiration, and had better pre-PCI rates of TIMI 2/3 flow but similar post-PCI rates of TIMI 3 flow.

As presented in S2 and S3 Tables, baseline characteristics, procedural data, and medications among patients randomized to heparin versus bivalirudin in the inter-hospital transfer and direct admission groups were well-matched.

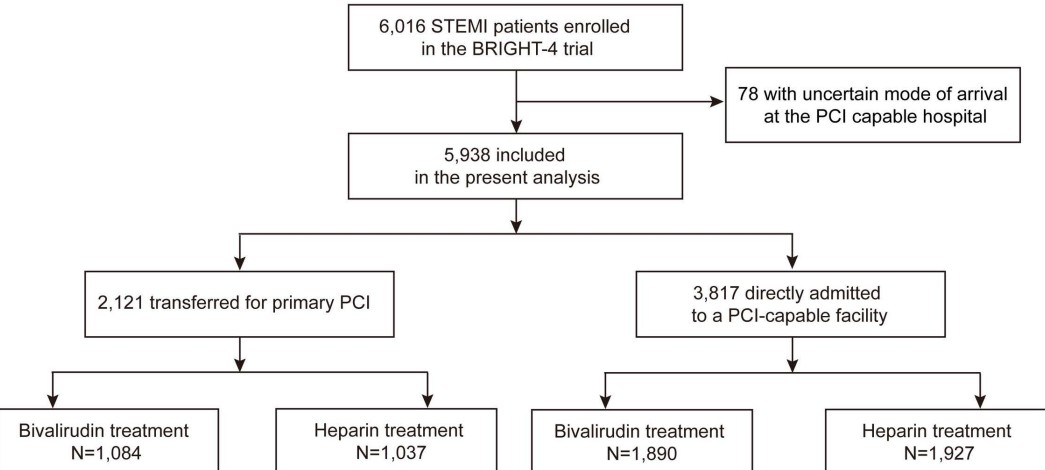

**Fig 1. The flowchart of the study population.** PCI, percutaneous coronary intervention; STEMI, ST-segment elevation myocardial infarction.

**Table 1. Baseline characteristics.**

| | Inter-hospital transfer (*N*=2,121) | Direct admission (*N*=3,817) | *P* value |
|---|---|---|---|
| Age, years | 60.3±12.6 | 60.7±11.9 | 0.33 |
| Male | 1,678 (79.1%) | 2,998 (78.5%) | 0.61 |
| Body mass index, kg/m$^2$ | 24.8±3.9 | 24.9±3.6 | 0.09 |
| Medical history | | | |
| Hypertension | 1,038 (48.9%) | 1993 (52.2%) | 0.02 |
| Diabetes mellitus | 475 (22.4%) | 862 (22.6%) | 0.87 |
| Smoking | | | 0.0001 |
| Active | 1,038 (48.9%) | 1,696 (44.4%) | |
| Former | 183 (8.6%) | 286 (7.5%) | |
| Never | 900 (42.4%) | 1835 (48.1%) | |
| Previous MI | 81 (3.8%) | 302 (7.9%) | <0.0001 |
| Previous PCI | 68 (3.2%) | 302 (7.9%) | <0.0001 |
| Previous stroke | 243 (11.5%) | 452 (11.8%) | 0.66 |
| Killip class | | | 0.74 |
| I | 1,299 (61.2%) | 2,339 (61.3%) | |
| II | 589 (27.8%) | 1,088 (28.5%) | |
| III | 162 (7.6%) | 278 (7.3%) | |
| IV | 71 (3.3%) | 112 (2.9%) | |
| Hemoglobin, g/dL | 139.0±18.0 | 140.9±18.2 | <0.0001 |
| Anemia* | 485 (22.9%) | 723 (18.9%) | 0.0003 |
| Platelet count, ×10$^9$/L | 225.6±69.5 | 230.0±68.6 | 0.02 |
| eGFR, ml/min/1.73 m$^2$ | 108.2±34.9 | 103.7±32.8 | <0.0001 |
| <60 ml/min/1.73 m$^2$ | 152 (7.2%) | 251 (6.6%) | 0.39 |
| Symptom onset-to-FMC, hrs | 1.76 (0.95–4.00) | 2.37 (1.08–4.97) | <0.0001 |
| Transfer time, first hospital arrival to tertiary hospital arrival, min | 132 (73–254) | – | – |
| Symptom onset-to-tertiary hospital arrival, hrs | 4.97 (3.00–9.00) | 2.50 (1.50–5.00) | <0.0001 |
| ≤12 h | 1506/1827 (82.4%) | 3023/3298 (91.7%) | <0.0001 |
| >12 h | 321/1827 (17.6%) | 275/3298 (8.3%) | |
| FMC-to-wire time, hrs | 3.57 (2.45–5.52) | 1.28 (0.98–1.83) | <0.0001 |
| Tertiary hospital door-to-wire time, hrs | 0.98 (0.77–1.42) | 1.22 (0.92–1.73) | <0.0001 |
| Symptom onset-to-wire time, hrs | 6.00 (3.97–10.17) | 3.93 (2.81–6.73) | <0.0001 |
| Procedure duration, min† | 30.0 (20.0–42.0) | 29.0 (20.0–41.0) | 0.65 |

Data are shown as n (%), mean±SD, or median (interquartile range). eGFR, estimated glomerular filtration rate; FMC, first medical contact (defined as arrival time at the first hospital); MI, myocardial infarction; PCI, percutaneous coronary intervention.

*Hemoglobin <13 g/dL in men and <12 g/dL in women.

†Defined as the time from guiding catheter insertion to its withdrawal.

## Clinical outcomes

At 30 days, there were no significant differences in the rates of the primary composite outcome, all-cause mortality, BARC types 3–5 bleeding, stent thrombosis, MACCE or NACE between the inter-hospital transfer and direct admission groups. However, acquired thrombocytopenia was slightly more common in the inter-hospital transfer group. These findings were similar after adjustment for differences in baseline characteristics, procedural data, and medications (Table 3).

The outcomes of bivalirudin plus a post-PCI high-dose infusion for 2–4 hrs versus heparin monotherapy on 30-day clinical outcomes in the inter-hospital transfer and direct admission groups, adjusted for differences in baseline covariates

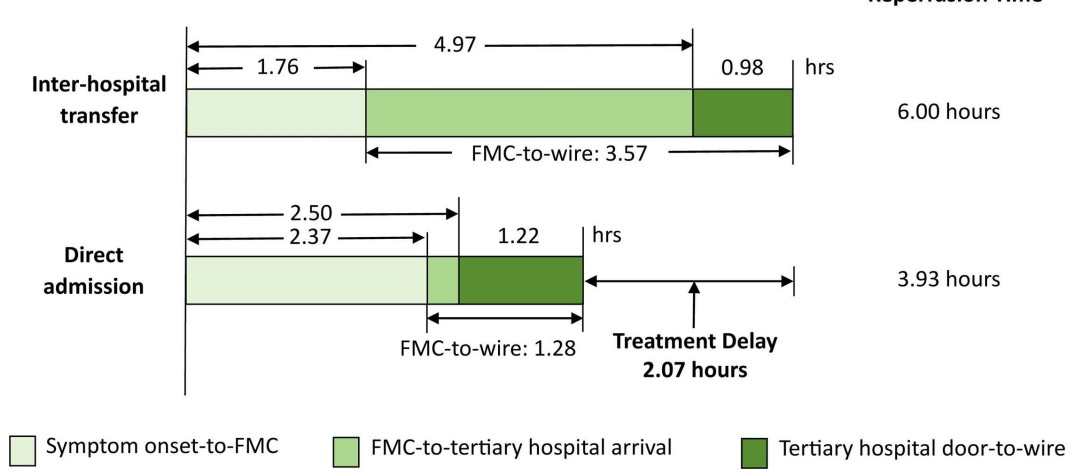

**Fig 2. Intervals for patients to a PCI facility.** PCI, percutaneous coronary intervention; FMC, first medical contact.

and treatments, are shown in Table 4. There were no significant interactions between randomized anticoagulation use and the transfer versus direct admission subgroup on any of the 30-day primary or secondary endpoints (Fig 3).

As shown in S4 Table, the results from both standard Cox models and Cox frailty models for the primary outcome were consistent, which suggested that the observed associations were not significantly influenced by inter-center variability. Furthermore, we conducted a PSM analysis, which identified 1882 well-balanced pairs. The results of this sensitivity analysis (see S5 Table) were in line with our primary findings. Lastly, we performed an additional analysis excluding patients with symptom onset-to-wire time exceeding 24 hrs, and applied multivariable Cox models. The results (see S6 and S7 Tables) remained consistent with those from the primary analysis.

### Analysis according to symptom onset-to-wire time

As shown in S8 and S9 Tables, significant differences emerged in 30-day outcomes between patients with a symptom onset-to-wire time of ≥3 hrs compared with <3 hrs. Delayed reperfusion was associated with greater 30-day cardiovascular, all-cause mortality and stroke, despite similar rates of reinfarction, stent thrombosis and major bleeding, and lower rates of ischemia-driven TVR. These findings were consistent in both the inter-hospital transfer and direct admission groups. According to S10 Table, we further stratified the symptom onset-to-wire time and found that the incidence of the primary outcome increased with longer symptom onset-to-wire intervals, regardless of whether the analysis was conducted in the overall population or in the directly admitted and transferred subgroups. Although the ≥12 hrs group showed a slight decrease in the primary outcome, this finding should be interpreted with caution given the relatively small number of patients in this category.

### Discussion

In the present analysis from the BRIGHT-4 trial, 30-day clinical outcomes were not significantly worse in patients with STEMI presenting at a non-PCI capable hospital who were transferred to a tertiary facility for primary PCI compared with direct admission at the interventional center, despite an increase in total ischemic time of approximately 2 hrs. In addition, the relative effects of bivalirudin with a post-PCI high-dose infusion for 2–4 hrs compared with heparin monotherapy, with GPI use reserved for bail-out complications in both arms, were consistent for all 30-day primary and secondary outcomes in the inter-hospital transfer and direct admission groups. The principal results from the BRIGHT-4 trial that bivalirudin

**Table 2. Medications and procedural data.**

| | Inter-hospital transfer (N=2,121) | Direct admission (N=3,817) | P value |
|---|---|---|---|
| **Medications** | | | |
| Study medications | | | |
| Heparin | 1,031 (48.7%) | 1922 (50.4%) | 0.21 |
| Total dose during PCI, U | 5,460 (4,550–6,550) | 5,600 (4,900–6,900) | 0.01 |
| Bivalirudin | 1,088 (51.3%) | 1894 (49.6%) | 0.21 |
| Post-PCI infusion administered | 1053/1053 (100.0%) | 1869/1869 (100.0%) | – |
| Post-PCI infusion duration, hrs | 3.0 (2.0–4.0) | 3.0 (2.3–4.0) | 0.003 |
| Additional bolus of study medications | 386 (18.2%) | 753 (19.7%) | 0.16 |
| Tirofiban for procedural thrombotic complications | 279 (13.2%) | 477 (12.5%) | 0.46 |
| Peak activated clotting time, sec | 290 (250-347) | 298 (256-351) | 0.003 |
| Dual antiplatelet therapy | | | |
| Aspirin | 2,113 (99.6%) | 3,788 (99.2%) | 0.07 |
| P2Y12 inhibitor | | | <0.0001 |
| Clopidogrel | 958 (45.2%) | 1,016 (26.6%) | |
| Ticagrelor | 1,163 (54.8%) | 2,801 (73.4%) | |
| **Procedural data** | | | |
| Arterial access* | | | 0.01 |
| Transradial | 1995/2119 (94.1%) | 3526/3816 (92.4%) | |
| Transfemoral | 124/2119 (5.9%) | 290/3816 (7.6%) | |
| Revascularization, any | 2068 (97.5%) | 3,774 (98.9%) | <0.0001 |
| Coronary arteries treated† | | | |
| Left main | 19 (0.9%) | 39 (1.0%) | 0.67 |
| Left anterior descending | 1,065 (51.5%) | 1807 (47.9%) | 0.008 |
| Left circumflex | 250 (12.1%) | 485 (12.9%) | 0.40 |
| Right | 817 (39.5%) | 1,603 (42.5%) | 0.02 |
| Multivessel intervention | 88 (4.3%) | 163 (4.3%) | 0.91 |
| PCI performed | 2059 (97.1%) | 3,761 (98.5%) | 0.0001 |
| Drug-eluting stent implantation | 1877 (88.5%) | 3,395 (88.9%) | 0.60 |
| Number of stents | 1.28±0.56 | 1.30±0.55 | 0.40 |
| Total length of stents, mm | 33.4±16.8 | 32.9±16.1 | 0.30 |
| Balloon angioplasty only | 182 (8.6%) | 366 (9.6%) | 0.20 |
| Thrombus aspiration | 313 (15.2%) | 746 (19.8%) | <0.0001 |
| TIMI flow, site-assessed | | | |
| Pre-PCI | | | 0.02 |
| 0/1 | 1,656 (81.0%) | 3,141 (83.9%) | |
| 2 | 164 (8.0%) | 257 (6.9%) | |
| 3 | 225 (11.0%) | 346 (9.2%) | |
| Post-PCI | | | 0.09 |
| 0/1 | 18 (0.9%) | 25 (0.7%) | |
| 2 | 33 (1.6%) | 38 (1.0%) | |
| 3 | 1996 (97.5%) | 3,684 (98.3%) | |
| Staged PCI within 30 days | 174 (8.2%) | 289 (7.6%) | 0.38 |
| Coronary artery bypass graft surgery | 9 (0.4%) | 13 (0.3%) | 0.61 |
| Coronary angiography only | 51 (2.4%) | 42 (1.1%) | 0.0001 |
| No angiography | 2 (0.1%) | 1 (0.03%) | 0.26 |

Data are shown as n (%), mean±SD, or median (interquartile range).

*Angiography was not performed in 1 patient in the direct admission group and 2 patients in the inter-hospital transfer group.

†Per patient; some patients had more than one epicardial coronary artery treated during the index percutaneous coronary intervention or bypass graft procedure, so the total is more than 100%. PCI, percutaneous coronary intervention. TIMI, thrombolysis in myocardial infarction.

**Table 3. Clinical outcomes at 30 days.**

| | Inter-hospital transfer (*N*=2,121) | Direct admission (*N*=3,817) | Unadjusted HR (95% CI) | *P* value | Adjusted HR (95% CI) | *P* value |
|---|---|---|---|---|---|---|
| Primary outcome: All-cause death or BARC types 3–5 bleeding | 88 (4.2%) | 131 (3.4%) | 1.21 (0.92, 1.59) | 0.17 | 0.99 (0.73, 1.33) | 0.94 |
| Death from any cause | 78 (3.7%) | 124 (3.2%) | 1.13 (0.85, 1.50) | 0.39 | 0.91 (0.66, 1.25) | 0.55 |
| From cardiovascular causes | 75 (3.5%) | 120 (3.1%) | 1.13 (0.84, 1.50) | 0.42 | 0.89 (0.64, 1.23) | 0.47 |
| BARC types 3–5 bleeding | 15 (0.7%) | 14 (0.4%) | 1.93 (0.93, 4.00) | 0.08 | 2.00 (0.93, 4.32) | 0.08 |
| Reinfarction | 18 (0.8%) | 24 (0.6%) | 1.35 (0.73, 2.49) | 0.34 | 1.41 (0.74, 2.68) | 0.29 |
| Ischemia-driven TVR | 8 (0.4%) | 19 (0.5%) | 0.76 (0.33, 1.73) | 0.51 | 0.82 (0.35, 1.92) | 0.65 |
| Stroke | 10 (0.5%) | 19 (0.5%) | 0.95 (0.44, 2.04) | 0.89 | 1.09 (0.50, 2.39) | 0.83 |
| Stent thrombosis | 17 (0.8%) | 27 (0.7%) | 1.13 (0.62, 2.08) | 0.69 | 1.20 (0.64, 2.25) | 0.57 |
| Acute (<24 hrs) | 8 (0.4%) | 10 (0.3%) | 1.44 (0.57, 3.65) | 0.44 | 1.39 (0.53, 3.65) | 0.50 |
| Subacute (1–30 days) | 9 (0.4%) | 17 (0.4%) | 0.95 (0.43, 2.14) | 0.91 | 1.10 (0.48, 2.54) | 0.82 |
| MACCE* | 102 (4.8%) | 171 (4.5%) | 1.07 (0.84, 1.37) | 0.57 | 0.97 (0.74, 1.27) | 0.83 |
| BARC types 2–5 bleeding | 41 (1.9%) | 99 (2.6%) | 0.74 (0.52, 1.07) | 0.11 | 0.79 (0.54, 1.16) | 0.23 |
| All-cause death or BARC types 2–5 bleeding | 113 (5.3%) | 212 (5.6%) | 0.96 (0.76, 1.20) | 0.70 | 0.87 (0.68, 1.12) | 0.29 |
| Acquired thrombocytopenia† | 100 (4.9%) | 120 (3.2%) | 1.54 (1.18, 2.00) | 0.002 | 1.41 (1.06, 1.86) | 0.02 |
| NACE‡ | 110 (5.2%) | 177 (4.6%) | 1.12 (0.88, 1.42) | 0.35 | 1.01 (0.78, 1.31) | 0.95 |

Event rates are number of events (Kaplan–Meier estimated percentages). BARC, Bleeding Academic Research Consortium. TVR, target vessel revascularization. MACCE, Major adverse cardiac or cerebral events. NACE, Net adverse clinical events.

*MACCE includes all-cause death, myocardial infarction, ischemia-driven target vessel revascularization, or stroke.

†Defined as nadir platelet count of <150×10⁹ cells/L after the index procedure in patients in whom the baseline platelet count was ≥150×10⁹ cells/L.

‡NACE includes MACCE or BARC types 3–5 bleeding.

reduced the 30-day composite outcome of all-cause death or major bleeding thus apply to patients both presenting at a PCI capable center or requiring transfer, despite the additional delay required.

Although transferring STEMI patients to a tertiary hospital for primary PCI prolongs the total ischemic time, previous studies evaluating the effect of inter-hospital transfer versus direct admission on mortality in patients with STEMI undergoing primary PCI have yielded conflicting results [3–6,8–11,16–19]. For example, Chan and colleagues reported from a single-center prospective registry study of 594 STEMI patients that inter-hospital transfer compared with direct admission to the PCI center was associated with a longer total median ischemic time of 78 min, as well as higher 30-day and 1-year mortality [16]. Conversely, in the randomized HORIZONS-AMI trial of 3,602 patients with STEMI, 30-day and 1-year clinical outcomes were comparable in patients transferred for primary PCI compared with those directly admitted to a tertiary center, despite a reperfusion delay of approximately 67 min [6]. However, in neither of these studies was primary

**Table 4. Clinical outcomes at 30 days according to randomization to bivalirudin vs. heparin.**

| | Direct admission (N=3,817) | | | Inter-hospital transfer (N=2,121) | | | P value for interaction |
|---|---|---|---|---|---|---|---|
| | Bivalirudin (N=1,890) | Heparin (N=1,927) | Adjusted HR (95% CI) | Bivalirudin (N=1,084) | Heparin (N=1,037) | Adjusted HR (95% CI) | |
| Primary outcome: All-cause death or BARC types 3–5 bleeding | 52 (2.8%) | 79 (4.1%) | 0.62 (0.43, 0.89) | 38 (3.5%) | 50 (4.8%) | 0.66 (0.42, 1.05) | 0.78 |
| Death from any cause | 50 (2.6%) | 74 (3.8%) | 0.64 (0.44, 0.92) | 37 (3.4%) | 41 (4.0%) | 0.79 (0.48, 1.30) | 0.44 |
| From cardiovascular causes | 50 (2.6%) | 70 (3.6%) | 0.67 (0.46, 0.98) | 35 (3.2%) | 40 (3.9%) | 0.77 (0.46, 1.28) | 0.64 |
| BARC types 3–5 bleeding | 3 (0.2%) | 11 (0.6%) | 0.26 (0.07, 0.98) | 2 (0.2%) | 13 (1.3%) | 0.11 (0.02, 0.53) | 0.52 |
| Reinfarction | 10 (0.5%) | 14 (0.7%) | 0.71 (0.32, 1.61) | 7 (0.6%) | 11 (1.1%) | 0.60 (0.21, 1.69) | 0.78 |
| Ischemia-driven TVR | 7 (0.4%) | 12 (0.6%) | 0.56 (0.22, 1.44) | 2 (0.2%) | 6 (0.6%) | 0.30 (0.05, 1.85) | 0.51 |
| Stroke | 9 (0.5%) | 10 (0.5%) | 0.95 (0.38, 2.35) | 6 (0.6%) | 4 (0.4%) | 1.41 (0.37, 5.39) | 0.57 |
| Stent thrombosis | 6 (0.3%) | 21 (1.1%) | 0.28 (0.11, 0.69) | 5 (0.5%) | 12 (1.2%) | 0.42 (0.14, 1.21) | 0.66 |
| Acute (<24 hrs) | 2 (0.1%) | 8 (0.4%) | 0.24 (0.05, 1.16) | 2 (0.2%) | 6 (0.6%) | 0.16 (0.02, 1.15) | 0.84 |
| Subacute (1–30 days) | 4 (0.2%) | 13 (0.7%) | 0.31 (0.10, 0.95) | 3 (0.3%) | 6 (0.6%) | 0.41 (0.09, 1.82) | 0.64 |
| MACCE* | 74 (3.9%) | 97 (5.0%) | 0.76 (0.56, 1.03) | 47 (4.3%) | 55 (5.3%) | 0.78 (0.51, 1.19) | 0.85 |
| BARC types 2–5 bleeding | 47 (2.5%) | 52 (2.7%) | 0.99 (0.66, 1.49) | 16 (1.5%) | 25 (2.4%) | 0.49 (0.25, 0.95) | 0.27 |
| All-cause death or BARC types 2–5 bleeding | 94 (5.0%) | 118 (6.1%) | 0.82 (0.62, 1.09) | 51 (4.7%) | 62 (6.0%) | 0.70 (0.47, 1.04) | 0.87 |
| Acquired thrombocytopenia† | 56 (3.0%) | 64 (3.4%) | 0.88 (0.61, 1.27) | 41 (3.9%) | 59 (5.9%) | 0.67 (0.45, 1.01) | 0.27 |
| NACE‡ | 76 (4.0%) | 101 (5.2%) | 0.75 (0.55, 1.01) | 47 (4.3%) | 63 (6.1%) | 0.67 (0.44, 1.00) | 0.75 |

Event rates are number of events (Kaplan–Meier estimated percentages). Abbreviations and footnotes as in Table 3.

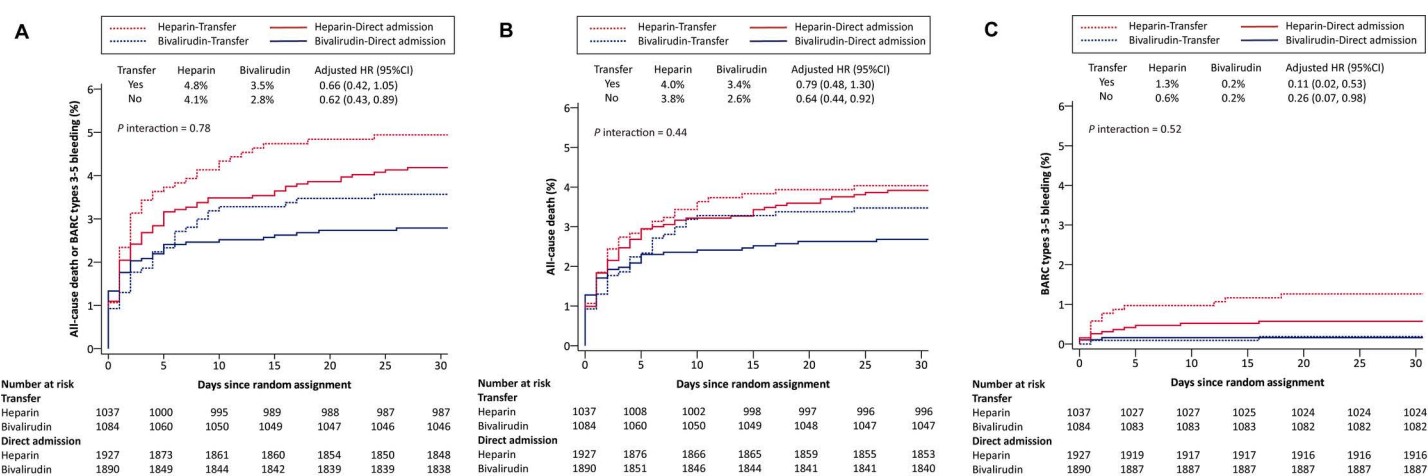

**Fig 3. Time-to-event curves at 30 days for the primary outcome and its components.** (A) All-cause death or BARC types 3–5 bleeding; (B) All-cause death; (C) BARC types 3–5 bleeding. BARC, Bleeding Academic Research Consortium.

PCI performed with the regimens that are most widely recommended and used in contemporary practice, namely heparin monotherapy and bivalirudin with a several hour post-PCI high-dose infusion. These were the randomized anticoagulation regimens used in the large-scale BRIGHT-4 trial. Moreover, in BRIGHT-4 primary PCI was also routinely performed with

radial artery access, and with use of ticagrelor in the majority of patients, practices that have been shown to further reduce mortality [20,21]. These considerations warranted a re-examination of the outcomes of STEMI patients presenting at non-PCI capable versus tertiary facilities in BRIGHT-4.

In BRIGHT-4, a substantial proportion (35.7%) of patients presenting with STEMI required inter-hospital transfer prior to primary PCI. This process delayed reperfusion by about 2 hrs. Nonetheless, after adjustment for differences in baseline covariates and treatments, this delay was not associated with a significant increase in mortality, major bleeding, MACCE or NACE. There might be several explanations for this observation. First, the benefit of reperfusion therapy in reducing mortality is greatest within the first 1–3 hrs after symptom onset, most likely due to the time-dependence of myocardial salvage before transmural infarction is established. After that early period, the impact of reperfusion therapy on reducing mortality is substantially less, and thus time to reperfusion is less critical [22,23]. In our study, 30-day cardiovascular and all-cause mortality was increased in patients with symptom onset-to-wire time of ≥3 hrs compared with <3 hrs, despite similar rates of ischemic and bleeding events, likely reflecting the impact of delayed reperfusion on myocardial salvage. Symptom onset-to-wire time ≥3 hrs was associated with mortality in both the direct admission and inter-hospital transfer groups. Thus, it is likely that the total ischemic time before reperfusion, rather than the mode of transfer, had a greater effect on survival in STEMI. In this regard, the somewhat shorter symptom onset to FMC and tertiary hospital door-to-wire times in the inter-hospital transfer group compared with the direct admission group likely mitigated the impact of the time required for inter-hospital transfer. Second, we cannot exclude the role that differences in baseline features and ancillary treatments in the inter-hospital transfer and direct admission groups might have had on the results. In the PL-ACS registry [4], transferred patients had higher-risk baseline characteristics, which might have contributed to their greater mortality at 30 days. In BRIGHT-4, however, clinical outcomes were similar both before and after covariate adjustment. Additionally, it is also worth noting that ticagrelor, providing faster and greater P2Y12 receptor inhibition than clopidogrel, was more frequently chosen in patients directly admitted for primary PCI, which might be due to a worse baseline clinical profile in the direct admission group or lack of ticagrelor in some referring hospitals. These limitations at referring hospitals emphasize the necessity for optimized reperfusion strategies in scenarios when delays are inevitable. Current guidelines advocate the pharmaco-invasive strategy when FMC-to-wire time exceeds 120 min [2]. However, our study excluded patients who received thrombolytic therapy prior to PCI, which limited our ability to evaluate the pharmaco-invasive strategy. This critical evidence gap highlights the urgent need for future studies comparing clinical outcomes between STEMI patients undergoing direct primary PCI and those receiving pharmaco-invasive therapy following inter-hospital transfer.

Importantly, in BRIGHT-4 the mode of transfer neither potentiated nor attenuated the benefits observed with bivalirudin with a post-PCI high-dose infusion for 2–4 hrs compared with heparin monotherapy. After covariate adjustment for differences in patient characteristics (e.g., differences in baseline TIMI flow rates) and adjunctive treatments (e.g., ticagrelor versus clopidogrel use, which take time to work in STEMI [24,25]), the relative hazards for all primary and secondary ischemic and bleeding outcomes with these anticoagulation regimens were consistent in the inter-hospital transfer and direct admission groups. Thus, the principal results of BRIGHT-4 apply to transfer patients as well as those presenting to a tertiary facility – STEMI patients undergoing primary PCI who are anticoagulated with bivalirudin plus a post-PCI high-dose infusion for 2–4 hrs have lower rates of all-cause mortality, major bleeding, stent thrombosis and NACE compared with heparin monotherapy [14]. Future studies with larger sample sizes or longer follow-up periods may help clarify whether this trend translates into a significant benefit.

The reduction in stent thrombosis in BRIGHT-4 after primary PCI with bivalirudin compared with heparin, even in patients presenting late after prolonged transfer, is notable. In the HORIZONS-AMI trial, the rate of stent thrombosis within the first 4 hrs after randomization was increased in patients treated with bivalirudin compared with heparin [26]. However, all heparin-treated patients in that trial also received a GPI, the only therapy shown to reduce stent thrombosis after stent implantation in STEMI [27], and bivalirudin was discontinued immediately after the PCI procedure, a risk factor for early stent thrombosis given its short (25-minute) half-life. In contrast, in BRIGHT-4, the platelet activation that is known to occur

after heparin use alone [28] likely resulted in increased rates of early stent thrombosis in patients randomized to heparin monotherapy without routine GPI administration, and in the bivalirudin arm, the median 3-hr post-PCI bivalirudin infusion was sufficient to suppress stent thrombosis in the early risk period, as first described in the EUROMAX trial [29].

Symptom onset-to-FMC time in BRIGHT-4 was longer than in previous studies, in part due to inclusion of patients with STEMI as late as 48 hrs after symptom onset, as opposed to 12 or 24 hrs in previous studies [3,4,6,17]. Moreover, the inter-hospital transfer group had a median FMC-to-wire time of 3.57 hrs (reflecting large inter-hospital distances in China, as well as other logistical considerations in transport), well past the guideline-recommended duration of 120 min [2]. As door-in door-out (DIDO) times of ≤30 min have been associated with lower in-hospital mortality [30], it is imperative to minimize referral-related delays. However, tertiary hospitals managed to significantly reduce door-to-wire times in transferred patients by proactively preparing the catheterization laboratory and bypassing the emergency room [6].

This study highlighted the value of a time-based stratified management strategy that prioritized total ischemic time over the specific mode of transfer. It underscored the essential role of timely reperfusion in optimizing outcomes for both directly admitted and transferred STEMI patients. The results supported the hypothesis that bivalirudin might serve as a preferable anticoagulant option during primary PCI, offering enhanced safety and survival benefits compared to heparin monotherapy, regardless of whether patients were transferred or not. Nonetheless, further research is warranted to evaluate the long-term prognostic implications of inter-hospital transfer and to assess the sustained benefits of bivalirudin therapy over time.

This study also has some limitations. First, as a *post hoc* analysis, randomization was not stratified by transfer status, and no adjustment was made for multiple comparisons, which may have increased the risk of type I error. Although multivariable analysis was used to adjust for differences in baseline characteristics, medications and procedures (most of which were modest) between the inter-hospital transfer and direct admission groups, selection bias and unmeasured confounders cannot be excluded. These results should be interpreted with caution in the context of an underpowered subgroup analysis. Accordingly, our results should be considered hypothesis-generating rather than conclusive inference and warrant dedicated, prospective confirmation. Second, we did not collect detailed data regarding the use of emergency medical services and DIDO times, resulting in our inability to accurately analyze all the reasons for treatment delay. Finally, as the present analysis reports only 30-day outcomes, further studies are needed to investigate the impact of transfer on the long-term prognosis of STEMI patients.

We concluded that in the BRIGHT-4 trial, transferring patients with STEMI presenting at a non-PCI capable hospital to a tertiary facility for primary PCI was not associated with significantly worse 30-day mortality, MACCE or NACE compared with direct admission, despite treatment delays and longer times to reperfusion. Procedural anticoagulation with a post-PCI high-dose bivalirudin infusion for 2–4 hrs was associated with lower rates of 30-day all-cause mortality, major bleeding, and stent thrombosis compared with heparin monotherapy, regardless of whether patients were transferred or admitted directly to the interventional facility for primary PCI.

## Supporting information

**S1 Table. Medications and procedural data.**
(DOCX)

**S2 Table. Baseline characteristics of patients randomized to heparin versus bivalirudin.**
(DOCX)

**S3 Table. Medications and procedural data in patients randomized to bivalirudin versus heparin.**
(DOCX)

**S4 Table. Comparison of HRs (95% CI) for the primary outcome: Cox model versus Cox frailty model.**
(DOCX)

**S5 Table. Clinical outcomes at 30 days after propensity score matching.**
(DOCX)

**S6 Table. Clinical outcomes at 30 days in patients with symptom onset-to-wire time ≤24 hrs.**
(DOCX)

**S7 Table. Clinical outcomes at 30 days according to randomization to bivalirudin versus heparin in patients with symptom onset-to-wire time ≤24 hrs.**
(DOCX)

**S8 Table. Clinical outcomes at 30 days according to symptom onset-to-wire time.**
(DOCX)

**S9 Table. Clinical outcomes at 30 days in the direct admission versus inter-hospital transfer groups according to symptom onset-to-wire time.**
(DOCX)

**S10 Table. Multivariable-adjusted HRs (95% CI) of 30-day primary outcome according to symptom onset-to-wire time.**
(DOCX)

**S1 Checklist. CONSORT 2025 checklist.**
(DOCX)

**S1 Text. Institutional review boards.**
(DOCX)

**S2 Text. Study protocol.**
(DOCX)

**S3 Text. Statistical analysis plan.**
(DOCX)

## Acknowledgments

The authors appreciate the dedicated efforts of all of the clinical research collaborators in the BRIGHT-4 trial organization.

## Author contributions

**Conceptualization:** Yaling Han, Gregg W. Stone.

**Data curation:** Miaohan Qiu, Zhenyang Liang, Yi Li, Yaling Han.

**Formal analysis:** Miaohan Qiu, Zhenyang Liang, Yi Li, Yaling Han.

**Funding acquisition:** Yaling Han.

**Investigation:** Xiaolin Su, Miaohan Qiu, Chengqi Gu, Xiuhui Yang, Bin Liu, Fanbo Meng, Bin Ning, Wei Li, Zhixiong Zhong, Zhengzhong Wang, Bei Shi, Zhuo Shang, Zhenyang Liang, Yi Li, Yaling Han.

**Methodology:** Xiaolin Su, Miaohan Qiu, Zhenyang Liang, Yi Li, Yaling Han.

**Project administration:** Yaling Han.

**Resources:** Yaling Han.

**Supervision:** Yaling Han.

**Validation:** Yaling Han.

**Writing – original draft:** Xiaolin Su.

**Writing – review & editing:** Miaohan Qiu, Chengqi Gu, Xiuhui Yang, Bin Liu, Fanbo Meng, Bin Ning, Wei Li, Zhixiong Zhong, Zhengzhong Wang, Bei Shi, Zhuo Shang, Zhenyang Liang, Yi Li, Yaling Han, Gregg W. Stone.

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
