## [Editor Report · Decision Letter 0]

Dear Dr Han,

Thank you for submitting your manuscript entitled "Impact of Inter-hospital Transfer on Clinical Outcomes After Primary PCI in STEMI: The BRIGHT-4 Trial" for consideration by PLOS Medicine.

Firstly, my sincere apologies for the delay in getting back to you. The holiday period and staff absences meant that it took much longer than anticipated to reach a final decision on some of the manuscripts submitted to us. Rest assured that this is not normally the case!

However, I am pleased to inform you that your manuscript has now completed its evaluation by the PLOS Medicine editorial team. I am writing to let you know that we would like to send your submission out for external peer review.

Please re-submit your manuscript within two working days, i.e. by Jan 09 2025 11:59PM. However, if this is not feasible, do please let us know and we can provide you with more time - feel free to email me directly (ssunny@plos.org).

Kind regards,

Syba

Syba Sunny, MBBS, MRes, FRCPath

Associate Editor

PLOS Medicine

ssunny@plos.org

---

## [Decision Letter · Decision Letter 1]

Dear Dr Han,

Many thanks for submitting your manuscript "Impact of Inter-hospital Transfer on Clinical Outcomes After Primary PCI in STEMI: The BRIGHT-4 Trial" (PMEDICINE-D-24-04173R1) to PLOS Medicine. The paper has been reviewed by subject experts and statisticians; their comments are included below and can also be accessed here: [LINK]

As you will see, the reviewers find the work of potential interest and have made suggestions to strengthen the findings. In particular, the referees request additional sensitivity analyses, including propensity score matching, and subgroup analyses, clarification of how missing data were handled and whether Cox proportional hazards assumptions were tested, and clarification of the tertiary hospitals and treatment regimens used, among other issues. After discussing the paper with the editorial team and an academic editor with relevant expertise, I'm pleased to invite you to revise the paper in response to the reviewers' comments. We also ask you to further discuss the potential relevance of the findings for clinical implementation or healthcare policy. We plan to send the revised paper to some or all of the original reviewers, and we cannot provide any guarantees at this stage regarding publication.

We ask that you submit your revision by May 27 2025 11:59PM. However, if this deadline is not feasible, please contact me by email, and we can discuss a suitable alternative.

Don't hesitate to contact me directly with any questions (afarrell@plos.org).

Best regards,

Alison

Alison Farrell, Ph.D.

Senior Editor

PLOS Medicine

afarrell@plos.org

Comments from the reviewers:

Reviewer #1: The BRIGHT 4 trial was a randomised trial comparing bivalirudin with heparin in more than 6000 patients with STEMI. Approximately 36% of patients required transfer from the hospital of admission to a tertiary PCI-capable hospital. Although the time from symptom onset to first medical contact was significantly shorter in the transfer group, transfer caused a delay of about 2.5 hours from symptom onset to final hospital arrival compared with those who did not require transfer. Despite this delay, the outcomes of patients who required transfer were not significantly different from those who did not. These are interesting data, although they contradict the common belief that "time is muscle". Apparently, the baseline characteristics of the patients do not suggest that the transfer group necessarily has a lower risk that compensates for the presumed disadvantage of delayed treatment. However, more information is needed to clarify the possible reasons for this finding:

1) Were the tertiary hospitals the same for both transferred and non-transferred patients? Were there differences between these hospitals in terms of primary PCI volume and guideline-based management strategy? It is noteworthy that ticagrelor was used significantly more often in the hospitals receiving non-transferred patients.

2) It has been hypothesised that the disadvantage of treatment delay is almost eliminated when the interval from symptom onset is greater than 4-5 hours (JAMA, 23 February 2005-Vol 293, No 8). The authors show the percentage of patients with an interval >12 hours. However, the information on treatment interval should be more detailed: e.g. proportion of patients between 3 and 6 hours, between 6 and 12 hours. I am aware of the problems attributed to analyses of subgroups of subgroups, but the exercise of looking at outcomes according to these subgroups might be informative.

3) The implications for the current guideline recommendations of a pharmaco-invasive approach for expected delays of 2 hours or more need to be discussed.

Reviewer #2: As the statistical reviewer I will focus on methods and reporting. Overall, this is a well conducted post-hoc analysis and also well reported. I have some points to raise.

Major

1) There is no mention of missing data and how that was handled. At the end of the statistical analysis section this needs to be discussed. If all data were complete, please state so. if not, why weren't multiple imputation approaches considered?

2) This is a multi-centre study. why weren't random effects (shared frailty) Cox models considered?

3) a sensitiivty analysis with propensity score matching (between patients with and without in-hospital transfer) would go a long way in offering reasssurance for the findings.

Minor

1) was the proportional hazards assumption assessed and how? if it does not stand, consider time-varying covariates.

Reviewer #3: The statistical methods used in this manuscript are generally appropriate for the study objectives and are well-executed. A few main points to consider:

* Please clarify whether Cox proportional hazards assumptions were tested.

* Specify whether covariates were prespecified or selected based on univariate associations.

* The handling of missing data is not described (e.g., complete case analysis, imputation) and should be reported.

* Multiple secondary endpoints and subgroup analyses are tested without adjustment; considering adjustment and acknowledge increased type I error risk.

* Consider providing sensitivity analyses (e.g., excluding very late presenters, >24h symptom onset-to-wire time) to confirm robustness.

Minor:

* Terms like "symptom onset-to-wire time" and "symptom onset-to-reperfusion time" are used interchangeably; recommend standardizing.

* Non-significant hazard ratios may still have clinically important ranges (e.g., HR 0.66 [0.42-1.05]); better discussion of possible clinical relevance is advised.

* Direct references to Supplementary Tables should be embedded earlier in the main Results text for easier navigation.

---

* Please upload any figures associated with your paper as individual TIF or EPS files with 300dpi resolution at resubmission; please read our figure guidelines for more information on our requirements: http://journals.plos.org/plosmedicine/s/figures. While revising your submission, please upload your figure files to the PACE digital diagnostic tool, https://pacev2.apexcovantage.com/. PACE helps ensure that figures meet PLOS requirements. To use PACE, you must first register as a user. Then, login and navigate to the UPLOAD tab, where you will find detailed instructions on how to use the tool. If you encounter any issues or have any questions when using PACE, please email us at PLOSMedicine@plos.org.

* [EDITOR: CHECK FINANCIAL DISCLOSURES, COI, DAS, AND ETHICS STATEMENTS AND INCLUDE ANY NECESSARY REQUESTS]

* Please ensure that the study is reported according to the [XXXX] guideline and include the completed [XXXX] checklist as Supporting Information. When completing the checklist, please use section and paragraph numbers, rather than page numbers. Please add the following statement, or similar, to the Methods: "This study is reported as per [XXXX] guideline (S1 Checklist)."

FIGURES AND TABLES

SUPPLEMENTARY MATERIAL

REFERENCES

[STUDY TYPE-SPECIFIC REQUESTS - DELETE SECTIONS AS NECESSARY]

RCTs [REFER TO RCT CHECKLIST AND MEETING NOTES FOR DETAILS TO ADD]

* PLOS Medicine requires that all trials be prospectively registered in one of registries recognized by WHO. Please ensure that study registration details are included in the Methods section.

* Please structure the Methods section using the following sub-headings: Study design and participants, Randomization and masking, Procedures, Outcomes, Statistical analysis.

* The following outcomes measures [ADD DETAILS AS NEEDED OR DELETE BULLET POINT] appear to differ between the submitted manuscript and the protocol [and/or trial registry]. Please clarify and explain all discrepancies between the paper and protocol. If the outcomes were not prespecified in the protocol, please define them in the Methods (Outcomes section) as post hoc and explain why they were added. Post-hoc comparisons should be presented as hypothesis generating rather than conclusive.

* Please ensure that all prespecified outcomes (primary, secondary, and exploratory) are listed in the Methods/Outcomes section and indicate whether there are outcomes that are not presented in the current report.

* Please specify the dates (Month Day, Year) during which study enrollment and follow up occurred.

* Please include absolute numbers wherever you report percentages; eg, n/N (%)

* Please present the safety data for the study including numbers of specific events and whether or not adverse events are thought to be related to treatment. AEs should be reported in the abstract, per CONSORT and CONSORT-Harms.

* Please complete the CONSORT checklist (https://www.equator-network.org/reporting-guidelines/consort/) and ensure that all components of CONSORT are present in the manuscript, including how randomization was performed, allocation concealment, blinding of intervention, definition of lost to follow-up, power statement. When completing the checklist, please use section and paragraph numbers, rather than page numbers.

* Please report your abstract according to CONSORT for abstracts, following the PLOS Medicine abstract structure (Background, Methods and Findings, Conclusions) https://www.equator-network.org/reporting-guidelines/consort-abstracts/

* If your trial had to undergo important modifications in response to extenuating circumstances, please complete the CONSERVE-CONSORT checklist and provide in your Supporting Information; (https://www.equator-network.org/reporting-guidelines/guidelines-for-reporting-trial-protocols-and-completed-trials-modified-due-to-the-covid-19-pandemic-and-other-extenuating-circumstances-the-conserve-2021-statement/). When completing the checklist, please use section and paragraph numbers, rather than page numbers.

* In keeping with our commitment to Open Science, please include the study protocol document and analysis plan (including any amendments) as Supporting Information to be published with the manuscript if accepted.

* Please note that PLOS Medicine requires prospective, public registration of a data sharing plan (as part of mandatory clinical trials registration) for all clinical trials that began enrollment on or after January 1, 2019, in accordance with ICMJE requirements.

OBSERVATIONAL STUDIES

* Abstract: Please include the study design, population and setting, number of participants, years during which the study took place (enrollment and follow up), length of follow up, and main outcome measures.

* Please ensure that the study is reported according to the STROBE (or appropriate STOBE extension) guideline (available from: https://www.equator-network.org/reporting-guidelines/strobe) and include the completed STROBE (or STROBE extension) checklist as Supporting Information. Please add the following statement, or similar, to the Methods: "This study is reported as per the Strengthening the Reporting of Observational Studies in Epidemiology (STROBE) guideline (S1 Checklist)." When completing the checklist, please use section and paragraph numbers, rather than page numbers.

* [FOR POPULATION HEALTH/REGISTRY STUDIES] Please ensure that the study is reported according to the RECORD guideline (available from https://www.record-statement.org) and include the completed checklist as Supporting Information. Please add the following statement, or similar, to the Methods: "This study is reported as per the Reporting of Studies Conducted using Observational Routinely-Collected Data (RECORD) guideline (S1 Checklist)." When completing the checklist, please use section and paragraph numbers, rather than page numbers.

* [FOR POPULATION HEALTH ESTIMATES] Please ensure that the study is reported according to the GATHER statement (available from https://www.equator-network.org/reporting-guidelines/gather-statement) and include the completed checklist as Supporting Information. Please add the following statement, or similar, to the Methods: "This study is reported as per the Guidelines for Accurate and Transparent Health Estimates Reporting (GATHER) statement (S1 Checklist)." When completing the checklist, please use section and paragraph numbers, rather than page numbers.

* [FOR MEDIATION ANALYSES] We recommend that the study is reported according to the AGReMA statement (https://agrema-statement.org/#:~:text=AGReMA%20is%20an%20evidence%2D%20and,randomised%20trials%20and%20observational%20studies) and include the completed checklist as Supporting Information. Please add the following statement, or similar, to the Methods: "This study is reported as per the Guideline for Reporting Mediation Analyses (AGReMA) statement (S1 Checklist)." When completing the checklist, please use section and paragraph numbers, rather than page numbers.

* For all observational studies, in the manuscript text, please indicate: (1) the specific hypotheses you intended to test, (2) the analytical methods by which you planned to test them, (3) the analyses you actually performed, and (4) when reported analyses differ from those that were planned, transparent explanations for differences that affect the reliability of the study's results. If a reported analysis was performed based on an interesting but unanticipated pattern in the data, please be clear that the analysis was data driven.

* Please state in the Methods section whether the study had a prospective protocol or analysis plan. If a prospective analysis plan (from your funding proposal, IRB or other ethics committee submission, study protocol, or other planning document written before analyzing the data) was used in designing the study, please include the relevant document(s) with your revised manuscript as a Supporting Information file to be published alongside your study and cite it in the Methods section. A legend for this file should be included at the end of your manuscript. If no such document exists, please make sure that the Methods section transparently describes when analyses were planned, and when/why any data-driven changes to analyses took place. Changes in the analysis, including those made in response to peer review comments, should be identified as such in the Methods section of the paper, with rationale.

MODELLING STUDIES

The following list is derived from Geoffrey P Garnett, Simon Cousens, Timothy B Hallett, Richard Steketee, Neff Walker. Mathematical models in the evaluation of health programmes. (2011) Lancet DOI:10.1016/S0140-6736(10)61505-X:

* If pertinent, please provide a diagram that shows the model structure, including how the natural history of the disease is represented, the process and determinants of disease acquisition, and how the putative intervention could affect the system.

* Please provide a complete list of model parameters, including clear and precise descriptions of the meaning of each parameter, together with the values or ranges for each, with justification or the primary source cited and important caveats about the use of these values noted.

* Please provide a clear statement about how the model was fitted to the data, including goodness-of-fit measure, the numerical algorithm used, which parameter varied, constraints imposed on parameter values, and starting conditions.

* For uncertainty analyses, please state the sources of uncertainties quantified and not quantified [can include parameter, data, and model structure].

* Please provide sensitivity analyses to identify which parameter values are most important in the model. Uncertainty estimates seek to derive a range of credible results on the basis of an exploration of the range of reasonable parameter values. The choice of method should be presented and justified.

* Please discuss the scientific rationale for the choice of model structure and identify points where this choice could influence conclusions drawn. Please also describe the strength of the scientific basis underlying the key model assumptions.

* For studies that develop a prediction model or evaluate its performance, please ensure that the study is reported according to the TRIPOD statement (https://www.equator-network.org/reporting-guidelines/tripod-statement) and include the completed checklist as Supporting Information. Please add the following statement, or similar, to the Methods: "This study is reported as per the Transparent Reporting of a Multivariable Prediction Model for Individual Prognosis Or Diagnosis (TRIPOD) statement (S1 Checklist)." For studies using machine learning, please use the TRIPOD-AI checklist. When completing the checklist, please use section and paragraph numbers, rather than page numbers.

DIAGNOSTIC STUDIES

* Please ensure that the study is reported according to the STARD guideline (https://www.equator-network.org/reporting-guidelines/stard/) and include the completed STARD checklist as Supporting Information. Please add the following statement, or similar, to the Methods: "This study is reported as per the Standards for Reporting of Diagnostic Accuracy (STARD) guideline (S1 Checklist)." When completing the checklist, please use section and paragraph numbers, rather than page numbers.

* Please structure your Abstract according to STARD for Abstracts (https://www.equator-network.org/reporting-guidelines/stard-abstracts/).

* Please structure the Methods section using the following sub-headings: Study design, Participants, Test methods, Analysis.

* Please include a diagram to describe the flow of participants through the study (typically figure 1).

MENDELIAN RANDOMIZATION STUDIES

* Please ensure that the study is reported according to the STROBE-MR guideline (https://www.equator-network.org/reporting-guidelines/strobe/) and include the completed STROBE-MR checklist as Supporting Information. Please add the following statement, or similar, to the Methods: "This study is reported as per the Strengthening the Reporting of Observational Studies in Epidemiology (STROBE) guideline, specific for mendelian randomization (S1 Checklist)." When completing the checklist, please use section and paragraph numbers, rather than page numbers.

* In the Introduction, please describe the exposure and the evidence for a potential causal relationship between exposure and outcome.

* In the Methods, please explicitly state the 3 core instrumental variable assumptions for the main analysis (relevance, independence, and exclusion restriction), as well assumptions for any additional or sensitivity analysis.

* In the Methods, please describe the MR estimator (e.g., 2-stage least squares, Wald ratio) and related statistics. Detail the included covariates and, in case of 2-sample MR, whether the same covariate set was used for adjustment in the 2 samples.

* If you are presenting an instrumental variable estimate, please compare this to the conventional observational estimate.

* Report the associations between genetic variant and exposure and between genetic variant and outcome, preferably on an interpretable scale.

* Report MR estimates of the relationship between exposure and outcome and the measures of uncertainty from the MR analysis, on an interpretable scale, such as odds ratio or relative risk per SD difference.

* If relevant, please consider translating estimates of relative risk into absolute risk for a meaningful time period.

* Please consider including plots to visualize results (e.g., forest plot, scatterplot of associations between genetic variants and outcome vs between genetic variants and exposure).

SURVEY-BASED STUDIES

* Please ensure that the study is reported according to the CROSS guideline (https://www.equator-network.org/reporting-guidelines/a-consensus-based-checklist-for-reporting-of-survey-studies-cross/) and include the completed CROSS checklist as Supporting Information. Please add the following statement, or similar, to the Methods: "This study is reported as per A Consensus-Based Checklist for Reporting of Survey Studies (CROSS) guideline (S1 Checklist)." When completing the checklist, please use section and paragraph numbers, rather than page numbers.

* Please report your survey response rates according to AAPOR recommendations (https://aapor.org/standards-and-ethics/best-practices/)

* Please define how the population surveyed was sampled.

* Please compare characteristics of respondents and nonrespondents if possible.

* If sequential waves of the survey were sent, please specify whether the characteristics of respondents changed over time or remained constant.

* Please include the survey response rate in the Abstract.

* Please include a copy of the survey in the supplementary files.

SYSTEMATIC REVIEWS & META-ANALYSES

* Please report your SR/MA according to the PRISMA guidelines provided at the EQUATOR site. http://www.equator-network.org/reporting-guidelines/prisma/. Please provide the completed PRISMA checklist as Supporting Information. When completing the checklist, please use section and paragraph numbers, rather than page numbers. Please add the following statement, or similar, to the Methods: "This study is reported as per the Preferred Reporting Items for Systematic Reviews and Meta-Analyses (PRISMA) guideline (S1 Checklist)."

* Abstract: Please report your abstract according to PRISMA for abstracts (https://doi.org/10.1371/journal.pmed.1001419) following the PLOS Medicine abstract structure (Background, Methods and Findings, Conclusions). Please ensure you provide dates of search, data sources, number of studies included, types of study designs included, eligibility criteria, and synthesis/appraisal methods.

* Please note that we expect searches to be updated to within 6 months of the time of submission.

QUALITATIVE STUDIES

* Please report your qualitative study according to the appropriate study design provided at (http://www.equator-network.org/?post_type=eq_guidelines&eq_guidelines_study_design=qualitative-research&eq_guidelines_clinical_specialty=0&eq_guidelines_report_section=0&s=) and provide the relevant completed checklist as a supplemental file. In the checklist, please include sufficient text excerpted from the manuscript to explain how you accomplished all applicable items. When completing checklists, please use section and paragraph numbers, rather than page numbers.

* We recommend that authors use the COREQ checklist, or other relevant checklists listed by the Equator Network, such as the SRQR, to ensure complete reporting (see: http://www.equator-network.org/?post_type=eq_guidelines&eq_guidelines_study_design=qualitative-research&eq_guidelines_clinical_specialty=0&eq_guidelines_report_section=0&s=). Please add the following statement, or similar, to the Methods: "This study is reported as per the Consolidated criteria for reporting qualitative research (COREQ): a 32-item checklist for interviews and focus groups (S1 Checklist)."

* In general, we expect qualitative studies to include the following: 1) defined objectives or research questions; 2) description of the sampling strategy, including rationale for the recruitment method, participant inclusion/exclusion criteria and the number of participants recruited; 3) detailed reporting of the data collection procedures; 4) data analysis procedures described in sufficient detail to enable replication; 5) a discussion of potential sources of bias; and 6) a discussion of limitations.

HEALTH ECONOMICS / COST-EFFECTIVENESS STUDIES

* Please ensure that the study is reported according to the CHEERS guideline (available from: https://www.equator-network.org/reporting-guidelines/cheers) and include the completed checklist as Supporting Information. Please add the following statement, or similar, to the Methods: "This study is reported as per the Strengthening the Consolidated Health Economic Evaluation Reporting Standards 2022 (CHEERS 2022) Statement (S1 Checklist)." When completing the checklist, please use section and paragraph numbers, rather than page numbers.

---

## [Decision Letter · Decision Letter 2]

Dear Dr. Han,

Thank you very much for re-submitting your manuscript "Impact of Inter-hospital Transfer on Clinical Outcomes After Primary PCI in STEMI: The BRIGHT-4 Trial" (PMEDICINE-D-24-04173R2) for review by PLOS Medicine.

I have discussed the paper with my colleagues and the academic editor and it was also seen again by three reviewers. I am pleased to say that provided the remaining editorial and production issues are dealt with we are planning to accept the paper for publication in the journal.

The remaining issues that need to be addressed are listed at the end of this email. We also ask that you revise the title in line with PLOS Medicine formatting to: "Analysis of inter-hospital transfer on clinical outcomes after primary percutaneous coronary intervention for ST-segment elevation myocardial infarction: A secondary analysis of the BRIGHT-4 trial".

Please also respond to the final comment of reviewer #2.

Any accompanying reviewer attachments can be seen via the link below. Please take these into account before resubmitting your manuscript:

[LINK]

We look forward to receiving the revised manuscript by Jul 01 2025 11:59PM.   

Sincerely,

Alison Farrell, Ph.D.

Senior Editor 

PLOS Medicine

plosmedicine.org

Requests from Editors:

* Please remove the 'conclusions' subheading from the discussion. Please also remove any other subheadings from the discussion.

* In the author summary, in the final bullet point of 'What Do These Findings Mean?', please include the main limitations of the study in non-technical language.

"* PLOS Medicine requires that the de-identified data underlying the specific results in a published article be made available, without restrictions on access, in a public repository or as Supporting Information at the time of article publication, provided it is legal and ethical to do so. Please see the policy at

http://journals.plos.org/plosmedicine/s/data-availability

and FAQs at

http://journals.plos.org/plosmedicine/s/data-availability#loc-faqs-for-data-policy "

"* The Data Availability Statement (DAS) requires revision. For each data source used in your study:

If the data are owned by a third party but freely available upon request, please note this and state the owner of the data set and contact information for data requests (web or email address).

*** Note that a study author cannot be the contact person for the data. Please confirm that the listed email address is not that of a study author.

* Did your study have a prospective protocol or analysis plan? Please state this (either way) early in the Methods section.

* Please remove any language that implies causality and refer to associations instead.

* Please confirm that your title complies with to PLOS Medicine's style. Your title must be nondeclarative and not a question. It should begin with main concept if possible. "Effect of" should be used only if causality can be inferred, i.e., for an RCT. Please place the study design ("A randomized controlled trial," "A retrospective study," "A modelling study," etc.) in the subtitle (ie, after a colon). Please see the requested revised title.

* Please ensure that all abbreviations are defined at first use throughout the text.

* Please define or remove BARC in the Abstract and explain onset-to-wire' for the general reader.

* Please identify by name all of the IRBs that approved the protocol.

* Please state in Methods any changes to the prespecified protocol for this analysis (if prespecified).

* Please identify the central illustration in the file. Is this figure 2? Is this illustration copyrighted (which is not permitted)?

* Please state in the Abstract where the trial was conducted and if multicenter.

* Please state in the Abstract the specific objectives/outcomes of this study.

* Does the "primary outcome" (Abstract) refer to the BRIGHT-4 trial or to the primary analysis in this study. Please clarify whether these are primary and/or secondary outcomes:30-day all-cause mortality, major bleeding and stent thrombosis.

Please use commas rather than dashes in confidence intervals.

Please do not refer to patients as admissions (line 65). Please use direct admission patients.

Line 98: Please remove causal language and use associative language.

Lne 105: does this statement refer to findings in this manuscript or in the BRIGHT-4 trial? If the latter, please remove or revise.

Please clarify for the general reader in the Abstract or Introduction what the high dose infusion is.

Please state in the Abstract and Introduction that this is a post-hoc secondary analysis, and indicate in Methods if prespecified.

Line 287: provide a figure callout, and not to 'central illustration'.

Please remove the Limitations subheading in the Discussion.

Comments from Reviewers:

Reviewer #1: No further comments

Reviewer #2: I am satisfied with the authors responses and the resulting additional analyses. This is not the caliper I would choose by the way (too large), and I would probably opt for a maximum of 0.05. I don't need to see the paper again.

Reviewer #3: Thanks. No further comments.

[LINK]

---

## [Editor Report · Decision Letter 3]

Dear Dr Han, 

On behalf of my colleagues and the Academic Editor, Andre-Pascal Kengne, I am pleased to inform you that we have agreed to publish your manuscript "Analysis of inter-hospital transfer on clinical outcomes after primary percutaneous coronary intervention for ST-segment elevation myocardial infarction: A secondary analysis of the BRIGHT-4 trial" (PMEDICINE-D-24-04173R3) in PLOS Medicine.

PRESS

Sincerely, 

Alison Farrell, Ph.D. 

Senior Editor 

PLOS Medicine